# Aluminum Nitride Piezoelectric Micromachined Ultrasound Transducer Arrays for Non-Invasive Monitoring of Radial Artery Stiffness

**DOI:** 10.3390/mi14030539

**Published:** 2023-02-25

**Authors:** Sheng Wu, Kangfu Liu, Wenjing Wang, Wei Li, Tao Wu, Heng Yang, Xinxin Li

**Affiliations:** 1State Key Laboratory of Transducer Technology, Shanghai Institute of Microsystem and Information Technology, Chinese Academy of Sciences, Shanghai 200050, China; 2School of Information Science and Technology, ShanghaiTech University, Shanghai 201210, China; 3School of Microelectronics, University of Chinese Academy of Sciences, Beijing 100049, China; 4East China Institute of Photo-Electron IC, Bengbu 233030, China

**Keywords:** piezoelectric micromachined ultrasound transducer array, pulse-echo method, miniaturized ultrasonic device, radial artery stiffness, blood pressure monitoring

## Abstract

An aluminum nitride (AlN) piezoelectric micromachined ultrasound transducer (PMUT) array was proposed and fabricated for non-invasive radial artery stiffness monitoring, which could be employed in human vascular health monitoring applications. Using surface micromachining techniques, four hexagonal PMUT arrays were fabricated within a chip area of 3 × 3 mm^2^. The mechanical displacement sensitivity and quality factor of a single PMUT were tested and found to be 24.47 nm/V at 5.94 MHz and 278 (in air), respectively. Underwater pulse-echo tests for the array demonstrated a −3 dB bandwidth of 0.76 MHz at 3.75 MHz and distance detection limit of approximately 25 mm. Using the PMUT array as an ultrasonic probe, the depth and diameter changes over cardiac cycles of the radial artery were measured to be approximately 3.8 mm and 0.23 mm, respectively. Combined with blood pressure calibration, the biomechanical parameters of the radial artery vessel were extracted using a one-dimensional vascular model. The cross-sectional distensibility, compliance, and stiffness index were determined to be 4.03 × 10^−3^/mmHg, 1.87 × 10^−2^ mm^2^/mmHg, and 5.25, respectively, consistent with the newest medical research. The continuous beat-to-beat blood pressure was also estimated using this model. This work demonstrated the potential of miniaturized PMUT devices for human vascular medical ultrasound applications.

## 1. Introduction

As one of the most powerful auxiliary tools for acquiring human health indices, ultrasound equipment has been developed for decades, having demonstrated the advantages of a real-time response, few side effects, and affordability. However, traditional ultrasonic diagnostic services tend to be accessible to patients only in the larger hospitals. Moreover, assessing brain, lung, or cardiovascular diseases, which require continuous ultrasound diagnosis, remains inconvenient. Consequently, the concept of point-of-care ultrasound (POCUS) has emerged, aimed at addressing specific pathological hypotheses in common healthcare scenarios [1,2]. It shows that portable ultrasonic devices with specific functions, from which people can acquire rapid healthcare diagnoses to improve their quality of life (even at home), are becoming popular. Based on the continued development of microscale ultrasonic probes, the ultrasound microelectromechanical system (MEMS) has been shown to have enormous potential. 

As distinct from traditional ultrasonic probes, micromachined ultrasonic transducers (MUTs) have established a new trend of low-power-consumption, high-level integration ultrasound applications, which can provide precise lateral resolution and handy frequency control design methods. Based on different mechanisms, capacitive micromachined ultrasonic transducers (CMUTs) utilize electrostatic force as the driving source, exhibiting the properties of large coupling effects and ultrahigh bandwidth [3,4]. In contrast, piezoelectric micromachined ultrasonic transducers (PMUTs) exhibit the advantages of a lower operating voltage, zero bias requirement, and linear output sensitivity, making them safe and convenient for use in medical ultrasonic applications. 

The acoustic performance of PMUTs is limited by the structure and material. To overcome the low electromechanical coupling of the flexural bending resonance mode, several solutions have emerged with the development of multielectrode configurations, bimorph structure, and array layouts [5,6,7]. As for the piezoelectric material of PMUTs, lead zirconate titanate (PZT) has been the most commonly used commercial piezoelectric material in integrated ultrasound detection systems [8,9,10]. However, its toxicity and environmentally harmful nature has pushed researchers to find lead-free piezoelectric material alternatives [11]. Aluminum nitride (AlN) offers the advantages of low permittivity and good thermal stability, making it an excellent choice [12]. Moreover, the IC-compatible fabrication process also makes AlN suitable for miniaturized ultrasound probe usage. 

With the help of ultrasound detection, the prediction and understanding of artery changes has an enormously important role in medical diagnostics, which can be useful in explaining the causes of cerebrovascular or cardiovascular diseases. Therefore, in previous related work, PMUT devices have been introduced into similar medical application scenarios. For example, Steve J. A. Majerus proposed an integrated CMOS front-end for wide-bandwidth interfacing of 0.8 mm polymer PMUT imaging catheters [13], Yande Peng proposed an invasive blood pressure monitoring method using AlN PMUTs on sheep [14], Hong Ding described a Doppler blood flowmeter using PMUTs in a flow-measurement mimic experiment [15], and Xiaoyue Jiang proposed a PZT PMUT array for radial artery motion tracking [16]. These experiments demonstrate promising applications of the PMUT array device in the field of vascular diagnosis. 

In this work, we validated and expanded the non-invasive continuous evaluation of the radial artery using AlN-based PMUT arrays, which could be used in microscale wearable devices. With the help of the surface micromachining MEMS process, a hexagonal PMUT array device was fabricated, which exhibited a high mechanical displacement sensitivity and quality factor. The acoustic performance of the array was examined by measuring the underwater resonance frequency, bandwidth, and distance detection capability using the pulse-echo method. Using the PMUT device as an ultrasound probe, the continuous dynamic changes of the radial artery were observed and recorded. Applying an artery model from previous studies [17,18], several medical indices including the arterial cross-sectional distensibility, compliance, and stiffness index were extracted by combining the dynamic radial artery changes with blood pressure values, which reflected the mechanical properties of the arterial vessel wall and could be used as indicators for artery stiffening in the clinical assessment. These parameters made it possible to detect vascular changes associated with vascular disease and to predict cardiovascular morbidity and mortality [19]. The extracted indices were found to match those of the latest medical research. Additionally, continuous blood pressure estimation over the cardiac cycle was conducted. The fabricated PMUT device developed had the advantages of a small contact area (with patients), applicability for continuous monitoring, and a safe, low operating voltage. Our study demonstrated the feasibility and potential of using PMUT arrays for continuous vascular health monitoring. In the future, MEMS technology for compact ultrasound biomedical chips for wearable devices could be realized. 

## 2. Design and Theory

### 2.1. PMUT Structure Design

The designed circular PMUT cell is shown in Figure 1a. We utilized the surface micromachining technique to build a diaphragm-on-cavity structure, which formed raised circular diaphragms over the wafer surface. Figure 1b shows a cross-sectional view of each material layer. 

Low-pressure chemical vapor deposition (LPCVD) polysilicon was used as the elastic layer due to its excellent conformality and uniformity. The thickness was determined to be 2 μm for high structural stability. Tetraethoxysilane (TEOS) SiO_2_ is used as the sacrificial layer due to the high etch selectivity between SiO_2_ and polysilicon in hydrofluoric acid. In addition, TEOS SiO_2_ could be deposited on both sides of the wafer with balanced stress and curvature. The sacrificial layer thickness determined the height of the cavity. The cavity sidewall provided a parasitic route for acoustic energy to leave the PMUT diaphragms and propagate into the substrate [20]. The higher the cavity, the lower the resonance energy leakage into the substrate. Hence, its thickness was determined to be 2 μm, which was the maximum amount in a single LPCVD procedure. The thermal dioxide thin layer provided high-quality electrical isolation with a thickness of 200 nm. Molybdenum was used as the metal electrodes and was determined to be as thin as 200 nm to lower the diaphragm rigidity. 

The piezoelectric coupling was calculated to find the proper thickness of the AlN film. The effective coupling coefficient was expressed and calculated as follows [20]:(1)$$k_{t}^{2}=\frac{{(4\pi \gamma \left(\gamma^{2}-1\right)\cdot e_{31,f}\cdot z_{p})}^{2}C_{m}}{C_{0}+{(4\pi \gamma \left(\gamma^{2}-1\right)\cdot e_{31,f}\cdot z_{p})}^{2}C_{m}}\;,$$
(2)$$C_{m}=\frac{{3r}^{2}}{64\pi D}{\;,\;C}_{0}=\frac{\varepsilon_{r}\varepsilon_{0}{\pi r}^{2}}{t_{AlN}}\;,\;$$
(3)$$D=\overset{N}{\underset{n=1}{\sum }}\frac{1}{3}Y_{11,n}^{\;}\cdot {\;(h}_{n}^{3}{-\;h}_{n-1}^{3})\;,$$
where *k_t_^2^* denotes the effective electromechanical coupling coefficient, *Cm* denotes the motional capacitor of the PMUT in resonance, *C*_0_ denotes the static capacitance of the PMUT, *D* denotes the flexural rigidity of the PMUT diaphragms, *γ* denotes the top electrode coverage on the radius and is designated as 0.7, *e*_31*,f*_ denotes the AlN film piezoelectric coefficient (taken as −1.08 C/m^2^), *z_p_* denotes the distance between the mid-plane of the piezoelectric layer and the neutral plane of the PMUT diaphragms, *r* denotes the radius of the PMUT circular diaphragm and can be eliminated in calculation, *ε_r_* denotes the relative permittivity in the AlN film (taken as 10), *ε_0_* denotes the vacuum permittivity, *t_AlN_* denotes the thickness of the AlN film, *Y*_11*,n*_ denotes the plate modulus of each layer, and *h_n_* denotes the distance between the top surface of each layer and the neutral plane. The calculation method of *z_p_*, *Y*_11*,n*_, *h_n_*, and the neutral plane have been discussed in detail in the literature [20] and are not further introduced here. 

The simulation results of the effective electromechanical coupling coefficient and flexural rigidity via MATLAB software (R2019b, MathWorks, Natick, MA, USA) are shown in Figure 2. On one hand, the flexural rigidity of the PMUT diaphragm increased with AlN film thickness. Large rigidity made it difficult to generate the volume change of the PMUT diaphragms, which lowered the mechanical sensitivity. On the other hand, the effective electromechanical coupling coefficient showed a peak value when the thickness of the AlN film was 1.2 μm. In our fabrication laboratory, the maximum thickness of the AlN film with a preferred orientation of (0002) was 1 μm. Varying the AlN film thickness from 0 to 1 μm, the coupling increased rapidly, and the rigidity changed slowly. Considering these factors, the thickness of the AlN was determined to be 1 μm. The thickness of each PMUT structure layer is shown in Table 1.

The finite element method (FEM) model was used to study the frequency attenuation of PMUTs in acoustic medium via COMSOL software (Version 5.3, COMSOL Co. Ltd., Burlington, MA, USA). As shown in Figure 3a, a PMUT with a radius of 43 μm was used; the eigenfrequency study showed that the fundamental resonance frequency was 6.47 MHz without the medium. The mechanical perfect matching layers (PML) formed the outer boundary with absorbing elements that ideally reflected no incident wave energy [20]. This method eliminated the rigid reflection of the substrate, and the absorbed energy was calculated as the anchor loss. The water was used as the acoustic medium and was set to cover the PMUT device. The acoustic PML of the water domain was used to eliminate acoustic reflection. As shown in Figure 3b, with an excitation voltage of 1 V, the absolute acoustic pressure at a distance of 2.8 mm on the axial direction changed with the PMUT excitation frequency. It showed that the underwater resonance frequency of a single PMUT was 3.58 MHz.

### 2.2. Hexagonal Array Design

As shown in Figure 4a, the parallel-connection hexagonal array was designed to improve acoustic performance. Traditional ultrasonic probes usually apply a square array of piezoelectric cells due to the size of the bulk material. 

By contrast, the MEMS fabrication technology provided a more flexible design approach for the array arrangement. As shown in Figure 4b, the simulation of the acoustic power along the azimuth angle was applied using the Phase Array System toolbox in MATLAB software (R2019b, MathWorks, Natick, MA, USA). With the same unit radius of 43 μm and the same pitch length of 119 μm, the acoustic power directional distribution comparison between a square array with 100 elements and a hexagonal array with 91 elements in a water environment was simulated. It showed that even with fewer PMUT units, the hexagonal array could produce almost the same amplitude output in the axial direction. The amplitude and number of acoustic side lobes were also smaller than those of the square array, making the hexagonal array more useful for acoustic focusing. In this work, a hexagonal array of 91 parallel PMUTs was used to generate a stronger acoustic focus. 

Additionally, the hexagonal array design also helped to lower down the impedance of the PMUT device, which was beneficial for constructing the interface circuits. Assuming all PMUT cells were identical, the total impedance of the array could be estimated by single cell impedance divided by the cell number. Figure 5a shows the modified Butterworth–Van Dyke (MBVD) equivalent circuit model of the PMUT array for extracting the electrical parameters from the COMSOL simulations (Version5.3, COMSOL Co. Ltd., Burlington, MA, USA). An initial approximation of the MBVD parameters was calculated from impedance simulations, and then each parameter was tuned by the relative error between the MBVD circuit calculation and the simulation results [21]. As shown in Figure 5b, with the array configuration, the impedance at resonance was simulated to be approximately 850 Ω with zero loads (no acoustic medium) and 2 kΩ with acoustic loads (in water medium). 

## 3. Fabrication Process

The PMUT surface micromachining processes are shown in Figure 6 and can be described as follows: After a standard wafer cleaning process, low-pressure chemical vapor deposition (LPCVD) TEOS-based silicon dioxide (2 μm) was deposited as the sacrificial layer. It was then patterned to form hexagonal arrays with circular cells using reactive ion etching (RIE). The etching sidewall profile angle of the sacrificial layer was approximately 70°, which helped the transferring of the piezoelectric and electrode films. After a standard wafer cleaning process, a LPCVD polysilicon layer (2 μm) was deposited on the TEOS film as the elastic layer. Annealing at 1050 °C in nitrogen was performed to reduce the stress and to improve the structure stability. Square microholes with a side length of 4 μm via polysilicon were etched using RIE for removing the sacrificial oxide. After wet etching in a 40% hydrofluoric acid solution, a circular diaphragm-on-cavity structure was formed. To avoid adhesion between the PMUT diaphragm and the substrate, the wafer was dried using supercritical fluid drying after the cleaning process. To seal the microholes on the polysilicon layer, a layer of LPCVD TEOS-based SiO_2_ was used for its excellent conformity and uniformity. A thin layer of SiO_2_ was generated inside the cavity simultaneously. After that, the undesirable SiO_2_ film outside the microholes on the wafer surface was removed by RIE. After a standard wafer cleaning process, thermal oxidation (0.2 μm) was performed to obtain an electrical insulation layer on top of the polysilicon. By using COMSOL simulation, the pressure difference on either side of the cavity exerted a small pressure bias on the PMUT diaphragm, which could lead to a small deflection and had little impact on the mechanical sensitivity. The thermal SiO_2_ layer induced compressive stress on the top surface, which could increase the deflection level. The compressive stress could lower the flexural rigidity, which helped to increase the mechanical sensitivity. After a standard wafer cleaning process, a 30 nm thick seed layer of AlN was sputtered on the wafer, which aimed for better orientation of molybdenum. Then molybdenum (0.2 μm) as the bottom electrode layer and AlN (1 μm) as the piezoelectric layer were sputtered sequentially, without breaking the vacuum environment. This process helped to obtain the preferred (0002) orientation of the AlN film. The top electrode film (0.2 μm) of the molybdenum was fabricated to the wafer using the lift-off process. Using photoresist as a mask, ion beam etching (IBE) was applied on the AlN film to form bottom electrode pads. For the convenience of wire bonding, a thin layer of gold (0.3 μm) was deposited on the pad using the lift-off process. 

## 4. Results: Fabrication and Testing

### 4.1. Fabricated PMUT Array 

The fabricated PMUT array chip comprised an area of 3 × 3 mm^2^, as shown in the scanning electron microscope (SEM) image in Figure 7a. Figure 7b shows an SEM image of a single PMUT structure. A cross-section of the cavity structure etched using a focused ion beam (FIB) method is shown in Figure 7c. As mentioned in the fabrication part, the deformation level at the center of the PMUT diaphragm was due to the pressure difference on both sides of the cavity and the stress in each material layer. By using a stylus profiler (Dektak XT, Bruker, Billerica, MA, USA), we tested the out-of-plane deformation level of 20 fabricated PMUT units separated in the same array; the average deflection at the center of the circular diaphragm was 0.582 μm, and the variation was 9.97%. The deformation levels of the PMUT units of 10 different arrays in and around the center of the wafer were also measured, and the variation was 17.35%. The variation was mainly due to film variations in each deposition step and etching variations in each etching step. By using a laser Doppler vibrometer (MSA-600, Polytec, Baden-Württemberg, Germany) with wideband signals, the individual PMUT resonance frequency was determined to be 5.94 MHz in the air. The error of the resonance frequency between the test results and the simulation results was 8.9%, and it could be influenced by the stress in each layer. Meanwhile, the quality factor was measured to be 278, which was automatically calculated within the laser Doppler vibrometer by dividing the resonant frequency by the −3 dB bandwidth. With the excitation of the sinusoidal signal at resonance frequency, the mechanical displacement sensitivity was measured to be 24.47 nm/V.

### 4.2. Pulse-Echo System and Performance Test

The fabricated PMUT chip was wire bonded to a customized PCB, before being covered with epoxy adhesive for wire protection as well as electrical isolation from the environment. Human silica gel (HY-E605, Hong Ye Jie Technology, Shenzhen, China) was then used to cover the device as an acoustic impedance matching layer, having low hardness and a similar density and acoustic speed to human tissue. Dynamic radial artery changes could be determined using the time-of-flight (TOF) method. As shown in Figure 8a, the excitation signal to the transmitting PMUT array was created via a signal generator (Agilent 33522A, Agilent Technologies, Santa Clara, CA, USA). 

Due to the acoustic impedance mismatch near the artery vessel wall, the ultrasound wave partly reflected, and the reflected signal was sensed by the receiving array. These echo signals were conditioned via a charge amplifier (VK10X, Vkinging, Shenzhen, China) and high-pass filter, before being displayed via an oscilloscope (TEKTRONIX MDO34, Beaverton, OR, USA). Then the data were recorded and analyzed using a computer. Figure 8b shows the gesture used for radial artery monitoring. Ultrasonic couplant (Kaidilan product, Wucheng, China) is used to eliminate the air gaps between the device and the skin. Figure 8c shows a size comparison of the PMUT probe device, indicating that the PMUT could have potential as a wearable device. 

The distance between the target and PMUT arrays via the TOF method can be expressed as follows:(4)$$d=c\;\cdot \;\frac{TOF}{2},$$
where *d* denotes the distance between PMUT array and artery wall, *c* denotes the sound velocity in human tissue (usually taken as 1540 m/s) [16], and *TOF* denotes the time difference between the excitation signal and the reflection signal.

The resonance frequency and bandwidth of the PMUT array were measured using the pulse-echo method in water since the acoustic impedance of human tissue is similar to that of water. By using a rigid reflector as target, the ultrasound generated from the transmitting array hit the reflector, then returned to the receiving array. Fixing the distance between the reflector and the PMUT device to be 7 mm, the frequency response of the echo signal amplitude is shown in Figure 9a. It showed that the resonance frequency was approximately 3.75 MHz (in water) with a −3 dB bandwidth of approximately 0.76 MHz. The error of the underwater resonance frequency between the test result and the simulation result shown in Figure 3b was within 4.8%. 

Three different distances between the reflector and the PMUT are set to test the distance detection capability via the TOF method. As shown in Figure 9b, with the excitation of five-cycle pulses at the resonance frequency, three test results are recorded and displayed (parts of the pink and blue lines are covered by the red one). Setting the distance as 4.5 mm (distance calibrated by micrometer with accuracy of 0.02 mm), the calculated distance was 4.62 mm using Equation 4, with an error of 2.67%. Setting the distance as 15 mm (pink line), the calculated distance was 14.53 mm, with an error of 3.1%. The enlarged view of the reflection signals (pink line) shows the ringing of the PMUT array. Setting the distance as 25 mm, the calculated result was 26.15 mm, with an error of 4.6%. The static distance detection error was mainly caused by the circuit delay of the signals and the selection of the signal boundary, which had little influence on the dynamic displacement detection. To test the transmitting and receiving sensitivity, the pressure at the reflector and the reflection pressure near the PMUT device were both measured by using a hydrophone (3290) at a distance of 15 mm. The transmitting sensitivity of array was calculated to be approximately 463 Pa/V. Using the COMSOL simulation shown in Figure 3a, the absolute pressure at 15 mm from one PMUT unit was 9.07 Pa/V. By multiplying the unit number, the simulated transmitting sensitivity of the array at 15 mm could be estimated to be 825 Pa/V. The receiving sensitivity of the array was calculated to be approximately 3.03 × 10^−2^ pC/kPa, when the feedback capacitance of the charge amplifier was 10pF and the magnification of the high-pass circuit was approximately 44. Using the COMSOL simulation, the receiving sensitivity of a single PMUT was 5.57 × 10^−4^ pC/kPa. By multiplying the unit number, the simulated receiving sensitivity of the array was 5.06 × 10^−2^ pC/kPa. The discrepancy between the simulation results and the test results was mainly due to the damping loss in the water medium and the material property variation, which were not considered in the simulation. In addition, due to the small size of the device, it was difficult to align the center axis of the PMUT array and the hydrophone, which could affect the transmitting sensitivity measurement. Similarly, the receiving sensitivity could be affected since the hydrophone could only be placed near the receiving array. Since the radial artery depth was less than 1 cm, the acoustic experiment showed that the PMUT device was capable of being used for superficial artery motion detection. 

In medical ultrasonic applications, the resolution of the pulse-echo method, which means the smallest size that can be distinguished from reflection signals, was expressed as follows [22]: (5)$$d_{r}=c\;\cdot \;\frac{T_{P}}{2},$$
where *d_r_* denotes the resolution in the pulse-echo test, and *T_P_* denotes the excitation pulse length. To minimize the resolution of the PMUT array and maintain a sufficient signal-to-noise ratio (SNR) using the TOF method, five-cycle excitation pulses at 3.75 MHz were used. The resolution could be determined to be 1.02 mm by Equation 5. It was sufficient to capture and distinguish the displacement of the two vessel walls of the radial artery since its diameter was approximately 3 mm. 

### 4.3. Radial Artery Stiffness Evaluation

The sectional algebraic pressure–area relationship of a one-dimensional model for the artery can be expressed as follows [17]:(6)$${P=P}_{ext}+\frac{{Eh}_{0}}{{(1-\nu }^{2})}\;\cdot \;\frac{\sqrt{\pi }}{A_{0}}\;\cdot \;(\;\sqrt{A}\;-\;\sqrt{A_{0}}\;)\;,$$
where *P* denotes the blood pressure of the vessel, *P_ext_* denotes the constant external pressure, *E* denotes Young’s modulus, *h*_0_ denotes the thickness of the vessel wall at the equilibrium state, ν denotes the Poisson’s ratio (typically taken to be 0.5), and *A* and *A*_0_ denote the cross-section area and that in the equilibrium state, which can be calculated by the artery diameter. 

By using a sphygmomanometer, the diastolic and systolic pressures on the left arm of the subject (27 years old and no vascular disease) were measured as 73 and 120 mmHg, respectively. The pulse pressure was calculated to be 47 mmHg. 

To monitor the dynamic change of the radial artery, the PMUT device was used on the radial artery of the left arm as shown in Figure 8b. The single acquisition signals were as shown in Figure 10a. 

The FFT frequency analysis of both the excitation and reflection signals after filtering is also displayed. The frequency consistency clarified that the reflection signal was caused by ultrasound echoes. In the enlarged view, two signal peaks reflected by the upper and lower vessel walls are easily distinguishable. Both the top and bottom envelopes were calculated and are displayed. The difference between the top and bottom envelopes was used to calculate the reflection signal amplitude. By extracting the reflection signal boundary, the depth of the radial artery was calculated to be 3.8 mm. 

With continuous recording, the three-dimensional reflection signal envelope amplitudes with calculated depth and recording time information are shown in Figure 10b, which represents the dynamic motion of the radial artery. Since the upper vessel wall of the radial artery was closer to the PMUT device, the reflection signal displayed a larger amplitude of approximately 40 mV peak to peak for the smaller attenuation. The second reflection signal from the lower vessel wall has a peak-to-peak value of approximately 20 mV. 

The movements of two reflection signal peaks reflected from the vessel walls were extracted via the TOF method. As shown in Figure 11, the motion of the upper vessel wall of the radial artery was smaller than that of the lower vessel wall. It was caused by the force exerted on the PMUT device when contacting the skin above the radial artery. The difference between the depths of the two walls represented the diameter change, as shown at the bottom of Figure 11. The average minimal diameter was 2.43 mm, while the average maximum value was 2.66 mm. Additionally, the heart rate was measured as 64 beats per minute. 

Combined with the systolic and diastolic blood pressure results, Equation 6 can be fitted. Moreover, the cross-sectional distensibility, compliance, and stiffness can be evaluated as follows [17,23]: (7)$$D=\frac{1}{A}\cdot \frac{dA}{dP}=\frac{{2\;(1-\nu }^{2})A_{0}}{{Eh}_{0}\sqrt{\pi A}}\;,$$
(8)$$C=\frac{dA}{dP}=\frac{{2\;(1-\nu }^{2}{)A}_{0}\sqrt{A}}{{Eh}_{0}\sqrt{\pi }}\;,$$
(9)$$STI=\frac{{ln\;(P}_{s}{/P}_{d})}{{(D}_{s}{-D}_{d}{)/D}_{d}}\;,$$
where *D* denotes the cross-sectional distensibility; *C* denotes the cross-sectional compliance; *STI* denotes the stiffness index; *P_s_* and *D_s_* denote the blood pressure and artery diameter in the systolic state, respectively; and *P_d_* and *D_d_* denote the blood pressure and artery diameter in the diastolic state, respectively. The calculation results and comparison with the previous medical literature are shown in Table 2. 

## 5. Discussion

The accuracy of artery stiffness calculations depends on the dynamic change monitoring of the radial artery. Using the PMUT device requires finger pressure on the PCB, which can lead to depth information signal jitter. The difference values of the upper and lower vessel wall depths, which represents the diameter change, are less effected by hand shaking and therefore more accurate. 

In the earlier literature, the measurements of the radial artery diameter change over cardiac cycles were often observed to be approximately 40 μm [29,30], leading to somewhat smaller cross-sectional distensibility and compliance values. In this study, the measured radial artery diameter change during the cardiac cycle was approximately 230 μm, a closer value to those of recent studies using high-resolution ultrasound methods [31,32]. 

In the latest research on arm arteries [27], the cross-sectional distensibility calculation methods are twice as small as those of our model in terms of mathematical expression; the measured data show good consistency with our measurement result (21.5%/100 mmHg vs. 4.03 × 10^−3^ mmHg^−1^). The cross-sectional compliance calculation method is close to our model in terms of mathematical expression; the measured data also show good consistency with our measurement result (1.3 × 10^−4^ cm^2^/mmHg vs. 1.87 × 10^−2^ mm^2^/mmHg). This comparison validates the feasibility and potential of monitoring artery health by using the AlN PMUT device. 

Additionally, the continuous beat-to-beat blood pressure was estimated with the radial artery diameter change and is shown in Figure 12, by using the fitted Equation (6). 

Since the PMUT device is pressed against the skin over the radial artery as shown in Figure 8c, hand shaking may cause small changes in the position of the PMUT device and thus affect the measurement results during long-term recording. If the PMUT device can be fixed on the skin over the radial artery, this signal noise can be eliminated. The mechanical properties of the artery vessel tissue change slowly, so this method may provide a portable solution for non-invasive blood pressure monitoring within days or hours. 

This work only used the blood pressure data of both systolic and diastolic states, so there is room for further improvement. If combined with a miniaturized pressure sensor, more data can be obtained for model fitting, making possible new evaluation and calibration systems for human vascular health monitoring. 

Since the piezoelectric coefficient of AlN is much lower than that of PZT, a non-invasive portable medical ultrasound probe using AlN material is rare. The practical medical ultrasound application with an individual PMUT unit is impossible. In this work, we applied the surface micromachining technique to fabricate the AlN PMUT structure with a low-pressure cavity. It reduced the squeeze-film damping and therefore increased the resonant quality factor and mechanical sensitivity. The hexagonal array configuration had greater wafer area utilization and larger concentrated acoustic performance than the traditional square array. Using a parallel-connection hexagonal array design, we integrated AlN PMUT units with high sensitivity to form an array with high acoustic performance in a small size (less than 1.5 × 1.5 mm^2^ for each 91-unit array). In addition, we demonstrated the first application of human superficial artery vessel stiffness monitoring and blood pressure estimation with the fabricated parallel-connection hexagonal AlN PMUT array. With further development, it can contribute to rapid self-health monitoring of hypertensive patients.

## 6. Conclusions

In this study, an AlN-based PMUT array device was proposed and fabricated for non-invasive ultrasonic monitoring of the radial artery. With the help of surface MEMS-based micromachining technology, multiple ultrasound transmitter arrays could be realized in an area of 3 × 3 mm^2^. Apart from the radial artery depth and diameter information, the artery stiffness indices were estimated via a one-dimensional artery model. The results were consistent with the latest medical studies. The continuous beat-to-beat blood pressure was also estimated via an established model. Our experiment demonstrated the feasibility and potential of minimized PMUT ultrasound devices. If further extended, this work could contribute to the fabrication of wearable medical ultrasonic devices of minimal size, paving the way toward point-of-care solutions for vascular health monitoring. 

## Figures and Tables

**Figure 1 micromachines-14-00539-f001:**
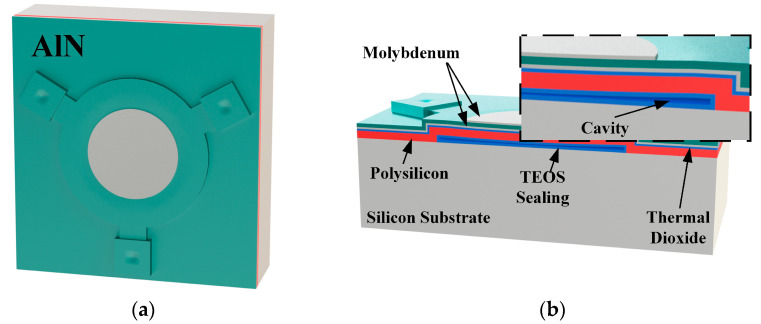
(**a**) The three-dimensional schematic of our PMUT unit. (**b**) The cross-section of a single PMUT unit and the enlarged view of the cavity structure.

**Figure 2 micromachines-14-00539-f002:**
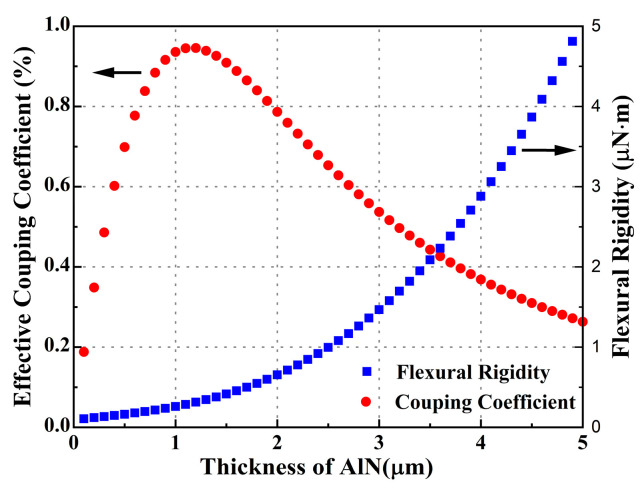
The relationship between the effective coupling coefficient and thickness of AlN film (red curve) and the relationship between the flexural rigidity and thickness of AlN film (blue curve).

**Figure 3 micromachines-14-00539-f003:**
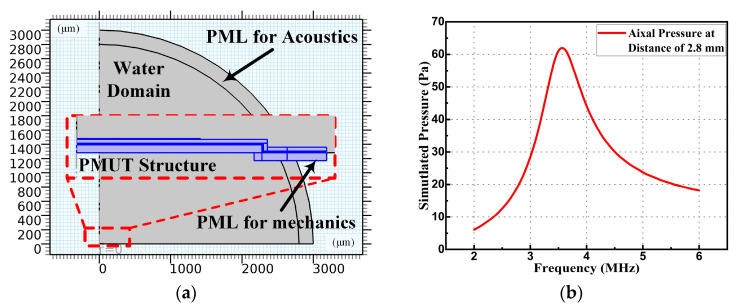
(**a**) Two-dimensional model of a PMUT with a perfect matching layer (PML) in COMOSL. (**b**) Frequency response of absolute acoustic pressure on the axial direction with the distance of 2.8 mm for one PMUT unit in water medium.

**Figure 4 micromachines-14-00539-f004:**
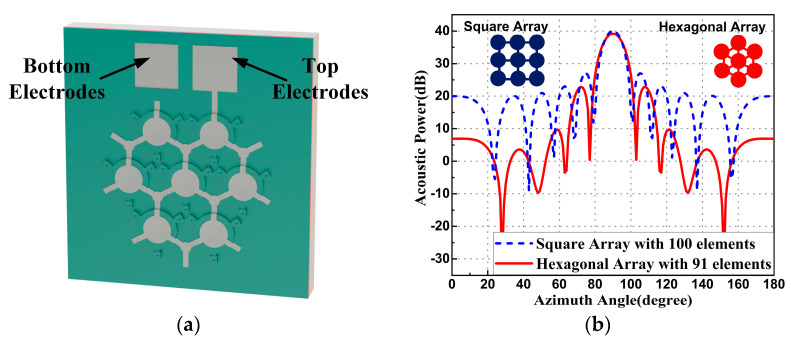
(**a**) Three-dimensional schematic view of a hexagonal PMUT array. (**b**) Acoustic power direction comparison between the square array and the hexagonal array.

**Figure 5 micromachines-14-00539-f005:**
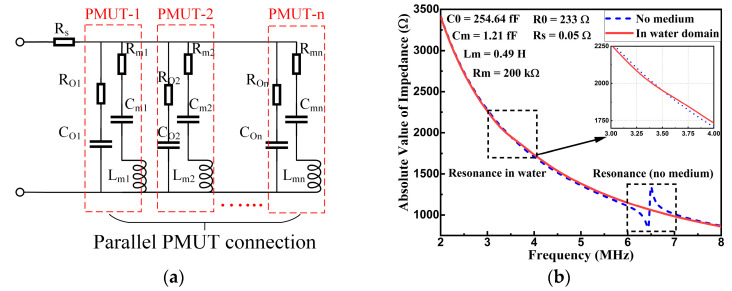
(**a**) The MBVD equivalent circuit model of the PMUT array. (**b**) The simulated frequency response of array impedance for both the no medium and water medium situations.

**Figure 6 micromachines-14-00539-f006:**
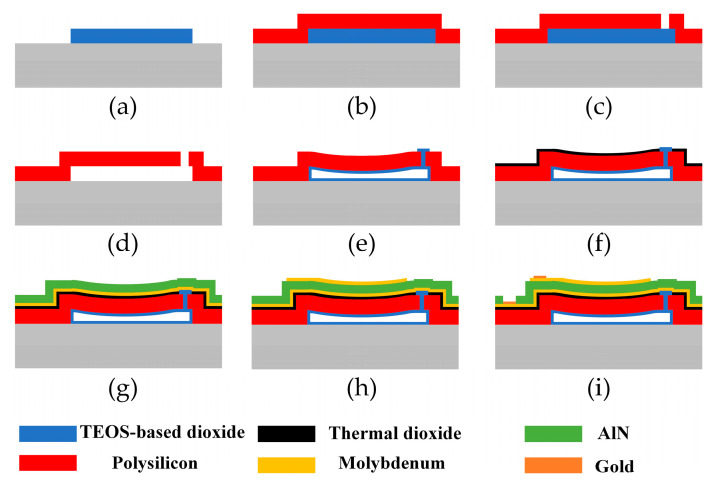
Schematic of PMUT fabrication process. (**a**) Deposition and patterning of sacrificial layer. (**b**) Deposition of elastic layer. (**c**) Etching of micro-holes. (**d**) Wet etching of sacrificial layer. (**e**) Sealing of the low-pressure cavity. (**f**) Thermal oxidation to form insulation layer. (**g**) Deposition of bottom electrodes and piezoelectric layer. (**h**) Deposition and patterning of top electrodes. (**i**) Deposition and patterning of gold layer on pads.

**Figure 7 micromachines-14-00539-f007:**
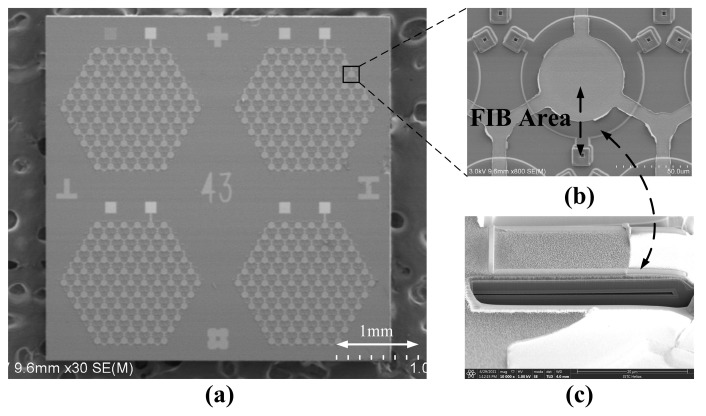
(**a**) SEM image of the fabricated PMUT chip with four arrays. (**b**) Enlarged view of a single PMUT unit. (**c**) Cross-sectional view of the PMUT structure via FIB cutting.

**Figure 8 micromachines-14-00539-f008:**
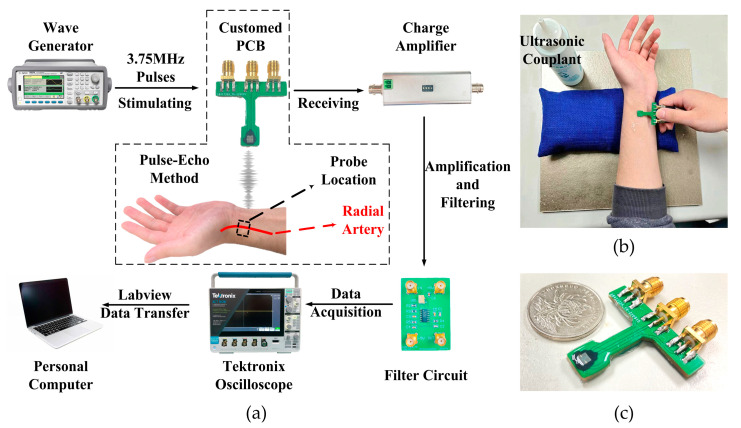
(**a**) Schematic of the pulse-echo detection signal transfer system. (**b**) Radial artery monitoring (corresponding to the dashed box in (**a**)). (**c**) Customized PCB for PMUT package and size comparison with a coin.

**Figure 9 micromachines-14-00539-f009:**
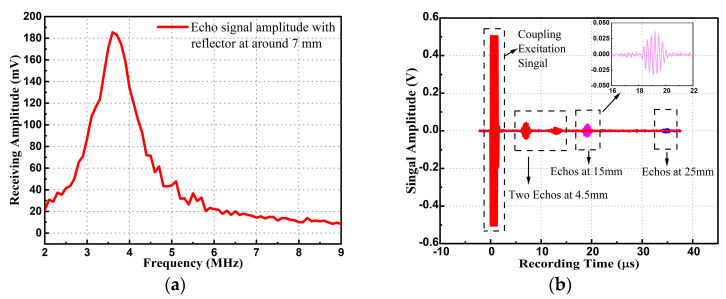
(**a**) Frequency response of underwater acoustic pulse-echo signals. (**b**) Pulse-echo signals with reflector at resonance frequency with three different detection distance.

**Figure 10 micromachines-14-00539-f010:**
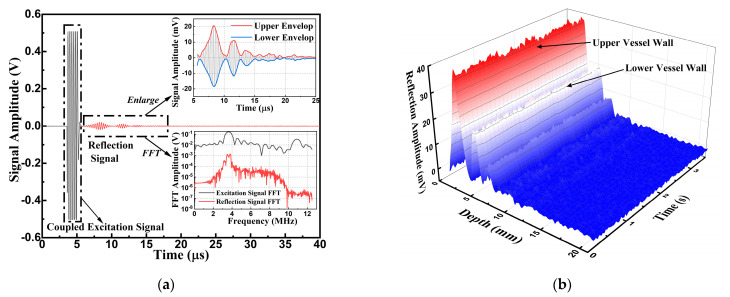
(**a**) Pulse-echo signals of the single acquisition and data process in MATLAB. (**b**) The three-dimensional view of the receiving signal amplitude of continuous recording.

**Figure 11 micromachines-14-00539-f011:**
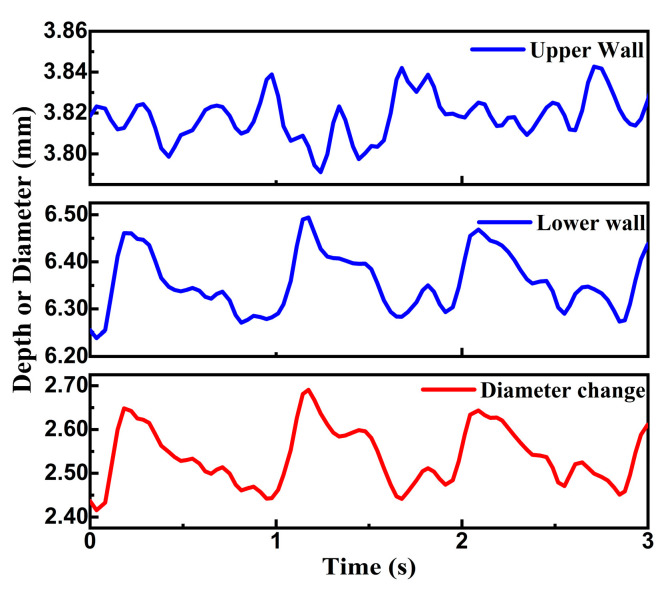
Dynamic depth change of two artery walls and diameter change.

**Figure 12 micromachines-14-00539-f012:**
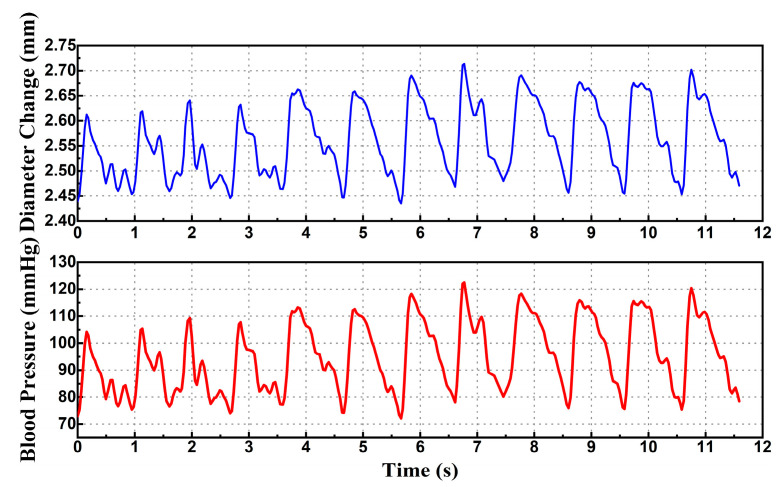
Continuous monitoring of radial artery diameter and estimated continuous beat-to-beat blood pressure.

**Table 1 micromachines-14-00539-t001:** Material and thickness for each layer of our PMUT structure.

Structure Layer	Material	Thickness
Sacrificial layer	TEOS	2 μm
Elastic layer	Polysilicon	2 μm
Insulation layer	Thermal dioxide	0.2 μm
Electrode layer	Molybdenum	0.2 μm
Piezoelectric layer	AlN	1 μm

**Table 2 micromachines-14-00539-t002:** Biomechanical parameters extracted from artery diameter change and blood pressure.

Biomechanical Parameters	This Work	Reference Values
Arterial distensibility	4.03 × 10^−3^/mmHg	~2 × 10^−3^/mmHg [24,25]
Arterial compliance	1.87 × 10^−2^ mm^2^/mmHg	~1.5 × 10^−2^ mm^2^/mmHg [26,27]
Stiffness index	5.25	4.41–8.83 [28]

## Data Availability

Data is unavailable due to privacy.

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
