# Peer review of "Aluminum Nitride Piezoelectric Micromachined Ultrasound Transducer Arrays for Non-Invasive Monitoring of Radial Artery Stiffness"

_micromachines, 2023, doi:10.3390/mi14030539_

Round 1

Reviewer 1 Report

Dear,

The paper demonstrates the use of pMUTs for biological applications related to radial arteries. Overall very nice work on the application side! However, at few places more explanation is needed, for example, in the introduction section the authors should introduce the significance of biological quantities such as stiffness index and so on- how they are useful for health monitoring and diagnosis. See the detailed comments (start with the line numbers referred to) for expected explanations and references. The paper can be accepted with these minor revisions.

36 Rephrase ‘ultrasonic diagnosis facilities’ with ‘Ultrasound Diagnostic Services

75 Rephrase ‘Attentional direction’

85 More explanation on the significance of biomechanical parameters of the radial artery vessel - cross‐sectional distensibility, compliance, and stiffness index is needed

101 Reference or explanation regarding energy leakage is needed

110 Capacitor or capacitance?

125 Use full form ‘laboratory’

138 More explanation on PML and anchor loss would improve the understandability, also  actuation voltage used for the axial pressure simulation should be stated

155 Is this a COMSOL simulation or any other software is used. An explanation on the array simulation environment and setup could be provided.

165 Provide reference and parameter extraction methodology could be described briefly.

183 & 187 Microhole size can be described.

213 Typo ‘measured’. Kindly check for typos and grammatical errors in the entire document.

246 For the underwater study, the transmitted pulse signal can be specified in more detail e.g., how many pulse cycles were used, etc. The magnified image of the echo signal would be useful to see the ringing of the pMUT. Moreover, insights into the gain of charge amplifier, pressure at the reflector, received pressure at pMUT, and transmit & receiving sensitivity would be helpful.  Also, compare the simulation results with experimental findings and discuss how closely it matches or the possible reasons for the mismatch.

Fig. 1&2- New sentence started with ‘And’

Reviewer 2 Report

The authors report their work on the design, simulation, and fabrication of an AlN PMUT device, and evaluated its possibility in the application of radial artery stiffness monitoring. The research background is comprehensively given, the device is fabricated with high quality, and the device testing process seems to be rigorous. Even though the novelty of the device itself is lacking, the potential in making it a portable medical device seems to be interesting:

1. Since a larger cavity height is preferred and the sacrifice layer thickness is limited by the LPCVD method, why didn’t the authors consider using a different silicon oxide deposition method (e.g. PECVD), or using a different material as the sacrifice layer?

2. A deformation of the cavity wall is observed in Fig. 6(e), after/during the cavity sealing step, what is the cause of this deformation and is there any consequential influence on the device performance? Is the deformation level in each of the units consistent throughout the whole device (is the deformation level the same among the 91 units)? What about the deformation level difference between different devices (wafer-to-wafer variation)?

3. Quality factor has its unique definition in different research fields, please clarify how the quality factor (page 7/13 line 213) is calculated in this work.

4. In Fig. 12, there is a clear trend of drifting up, in both the diameter change curve and the blood pressure curve. What is the reason for that? Besides, only 7 seconds of measurement data were provided, please provide longer measurement data to see if the drifting-up eventually stops or not.

5. The novelty is not clear. The hexagonal PMUT array has been reported before, please specify what was not possible before and what are the benefits. 

Round 2

Reviewer 2 Report

All the questions from the first round of review have been answered.